# Factors that mediate the relationships between household socio-economic status and childhood Attention Deficit Hyperactivity Disorder (ADHD) in children and adolescents: A systematic review

**Wolfgang A. Markham**[ID]*, **Nicholas Spencer**

Division of Health Sciences, Warwick Medical School, University of Warwick, Coventry, United Kingdom

* wolfgang.markham@warwick.ac.uk

## Abstract

### Background

ADHD is one of the most prevalent mental health disorders among children and adolescents. Household socio-economic status (SES) in early childhood is inversely related to ADHD later in childhood or adolescence. We conducted a systematic review to examine psychological, social and behavioural factors that mediate these relationships (PROSPERO Registration number: CRD42020182832).

### Methods and findings

We searched Medline, EMBASE, PsychINFo, and Web of Science from inception until May 2020. Both authors independently reviewed abstracts and identified papers for inclusion. We sought primary observational studies (cohort, cross-sectional and case control studies) of general population-based samples of children and adolescents aged 18 and under that investigated potential mediators of the relationships between SES and ADHD. Studies based upon non-general population-based samples, twins or biochemical/physiological changes were excluded. Direct and indirect effects derived from standard validated mediation analysis were extracted for potential mediators. We assessed risk of bias using a modified NIH tool and synthesised quantitative data without meta-analysis according to the (SWiM) protocol because of heterogeneity between included studies.

Family adversity, paternal and maternal ADHD symptoms, Home Learning Environment, breastfeeding duration and a combined fine motor and language score at age 2 may lie on the SES-ADHD pathway. Evidence concerning the influence of maternal depression/anxiety and adverse parenting was inconsistent across studies. There was no evidence that mother's health-related behaviour, family characteristics, child's consumption of fizzy drinks or other developmental characteristics at birth/during infancy lie on the SES-ADHD pathway. Publication bias may have been introduced by our decision not to search grey literature, not to approach study authors and limit the search to the English language.

**Data Availability Statement:** The search strategy used to identify the papers that were included in the systematic review is included as a

Supplementary file. The papers identified and included in the systematic review constitute the raw data that were used. All the data that were identified as relevant in the selected papers are included in the tables and figure within the paper or in the supplementary material.

**Funding:** The author(s) received no specific funding for this work.

**Competing interests:** The authors have declared that no competing interests exist.

## Conclusions

Evidence for mediation of the SES-ADHD pathway in childhood/adolescence is under-researched. Maternal mental health, family adversity, parenting and health-related behaviours warrant further research based on longitudinal data and employing the most advanced mediation analysis methods.

## Introduction

Attention Deficit Hyperactivity Disorder (ADHD) is among the commonest mental health disorders in childhood. The prevalence of children diagnosed with ADHD increased in the USA between 2003 and 2011 [1] and more children were in receipt of prescription drugs for the condition in the United States of America (USA) and western Europe in 2012 compared with 2005/6 [2]. Whether these trends reflect a true increase in prevalence or improved methods of data collection and case ascertainment remains unclear [3,4]. A meta-regression of over 100 studies, conducted in 2017, spanning the globe identified that prevalence rates hover around 5% [5].

The aetiology of ADHD is complex resulting from a range of biological, psychological and social conditions that can act individually or synergistically [6]. The association with household socioeconomic status (SES) is well-established [7]; however, explanations for the association vary from social conditions as causal [8], through reverse causality due to loss of earnings and relationship instability [9] to confounding by genetic factors that play a part in the aetiology of ADHD which may influence SES in indirect ways [10]. In considering the potential causal role of SES, a range of socially related prenatal, perinatal and early childhood risk factors probably interacting with genetic influences have been identified [11].

Russell AE et al. [12] suggest that these socially related risk factors may be on the causal pathway from SES to ADHD and mediate the relationship. Mediators are associated with both the exposure (SES) and the outcome (ADHD) and intervene between them accounting for some or all of the effect of the exposure on the outcome (Fig 1). Confounding variables, while causally related with both exposure and outcome [13], do not lie on the causal pathway between the exposure and the outcome but falsely obscure or accentuate the exposure/outcome relationship [14]. Moreover, treating mediators as confounding variables in regression models may also falsely lead to attenuation or elimination of the effect of the exposure on the outcome. Co-variates are related to the outcome but not the exposure and adjustment aims to improve the precision of the effect estimate [15]. In studies with SES as the exposure variable, adjusting in regression analysis for psychological, social or behaviour related variables which are potential mediators is likely to reduce the direct effect of SES on ADHD [16].

Mediation analysis aims to distinguish the total effect of the exposure on the outcome, the indirect effect of the potential mediator and the direct effect of the exposure. Mediation analysis is illustrated in Fig 1 where a = β coefficient of the path from exposure E to mediator M, b = β coefficient of the path from mediator M to outcome O, and c' = β coefficient of the path from the exposure E to outcome O. The direct effect of exposure E on outcome O is represented by c' and the indirect effect of mediator M on the path from exposure E to outcome O is the product of the unstandardised or standardised coefficients a x b [17].

This systematic review aims to assess published evidence from cohort, cross-sectional and case control studies regarding factors that mediate the SES-ADHD pathway in childhood and adolescence.

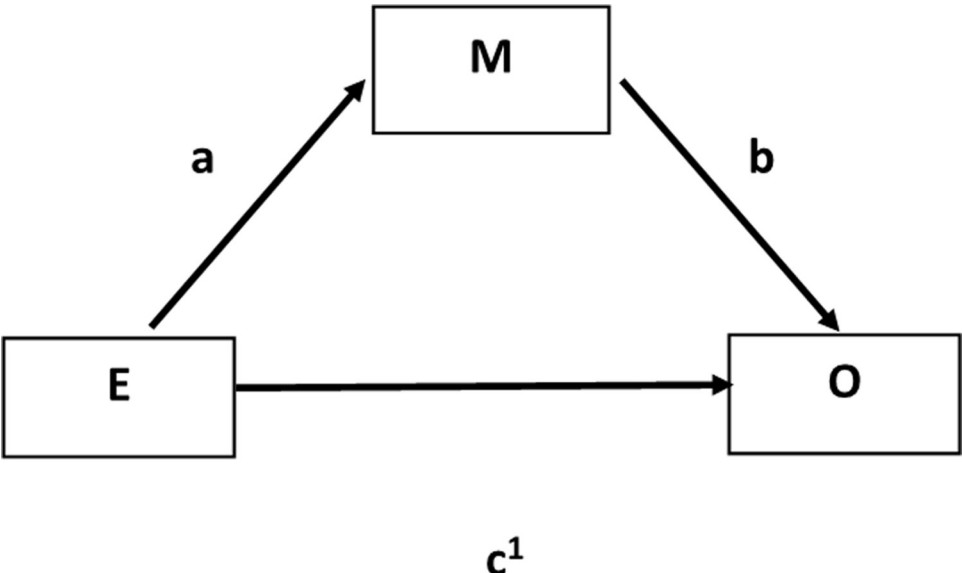

**Fig 1. Diagrammatic representation of mediation showing direct and indirect effects.**

## Methods

### Protocol registration and reporting

We conducted a systematic review according to a protocol that was registered in PROSPERO, an open access registry (registration number: CRD42020182832) (S1 File) [18]. We followed the PRISMA checklist [19] (S2 File) and SWiM (Synthesis Without Meta-Analysis) reporting guidelines [20] when reporting our findings.

### Search strategy and selection criteria

A health science librarian developed a search strategy to identify eligible investigations of mediators between household SES in early childhood and ADHD or high scores for hyperactivity/inattention in standard psychometric tests later in childhood or adolescence. We systematically searched Medline, EMBASE, PsychINFo and Web of Science from inception until 1st May 2020. A combination of indexed terms, free text words and MeSH headings were used (S3 File). We also manually searched reference lists of papers identified in the electronic database search including reviews. We did not search grey literature and study investigators were not contacted for unreported data or additional details. Included studies were restricted to the English language and were deduplicated. Both authors independently assessed article abstracts and full texts of studies that passed the initial screening phase. We included primary observational studies (cohort, cross-sectional and case control studies) of general population-based samples of children and adolescents under 19 years of age that used a recognised method for assessing mediation to investigate any psychological, social or behavioural factor that potentially mediated the relationship between household SES and childhood/adolescent ADHD or childhood/adolescent hyperactivity/inattention disorder assessed using standard psychometric tests. We excluded reviews and studies that were based upon non-general population samples, twins, adults over 18 years and studies where the investigated mediating factor was biochemical or physiological e.g. brain morphology or the investigated outcome was externalising behaviour including conduct problems or conduct disorder alone. At each stage disagreements between reviewers were resolved by consensus.

## Data analysis

Both authors independently extracted the following information from each included study—country, study type, population, sample size, attrition (%), SES, child's/adolescent's age at SES measurement, ADHD or hyperactivity/inattention measure and prevalence, child's/adolescent's age at ADHD measurement, mediators studied, mediation analysis method, covariates/confounders included in analyses, direct effects of SES and indirect effects of mediators (pathway coefficients), significant mediators, and mediators that were not significant. For each study, we estimated the proportion mediated derived by dividing indirect effects of mediators by the total effect of SES (indirect + direct effect) on ADHD expressed as a percentage [21]. Investigated psychological, social or behavioural factors that were identified as not meeting criteria for potential mediation because they were not significantly associated with either the exposure or the outcome were also identified. We then independently used a modified version of the NIH assessment tool for observational, cohort and cross-sectional studies [22] to assess the risk of bias (RoB) based on the quality of each included study's methodology (S1 Table). Additional questions were added to this NIH assessment tool namely:

- Was the study population representative of the whole target population?

- Was the mediation analysis clearly specified and defined?

- Was the choice of mediators clearly specified and justified?

- Were results of the mediation analyses clearly presented allowing direct and indirect effects to be distinguished?

  Studies were allocated high, moderate or low RoB based on methodological criteria (S1 Table). We synthesised quantitative data without meta-analysis according to the (SWiM) protocol [20] because of anticipated and confirmed sources of diversity and thus, heterogeneity between studies. These anticipated sources of diversity were:

- statistical diversity i.e. diversity in the methods for identifying the direct and indirect effects of household SES on ADHD and hyperactivity/inattention.

- methodological diversity i.e. included primary studies may be longitudinal studies, cross-sectional studies or case control studies.

- clinical diversity i.e. the outcome could be based upon a medical diagnosis of ADHD or having scores equal to or over the accepted cut-off point for hyperactivity/inattention on standard psychometric tests as reported by doctor, teacher, parent or self-reported.

- diversity in the measures of household SES i.e. based upon income, education, and other accepted measures of SES.

- diversity in the mediating factors investigated

- diversity in the measurement of potential mediators

- diversity in child's/adolescent's age at outcome measurement.

  The prioritisation of results was informal and based upon RoB assessments i.e. low RoB studies are prioritized over other studies and study design i.e. cohort studies are prioritized over other studies.

## Results

We identified n = 1130 citations from bibliographic databases and n = 5 citations from reference lists. After removing duplicates we screened n = 626 titles and abstracts for eligibility.

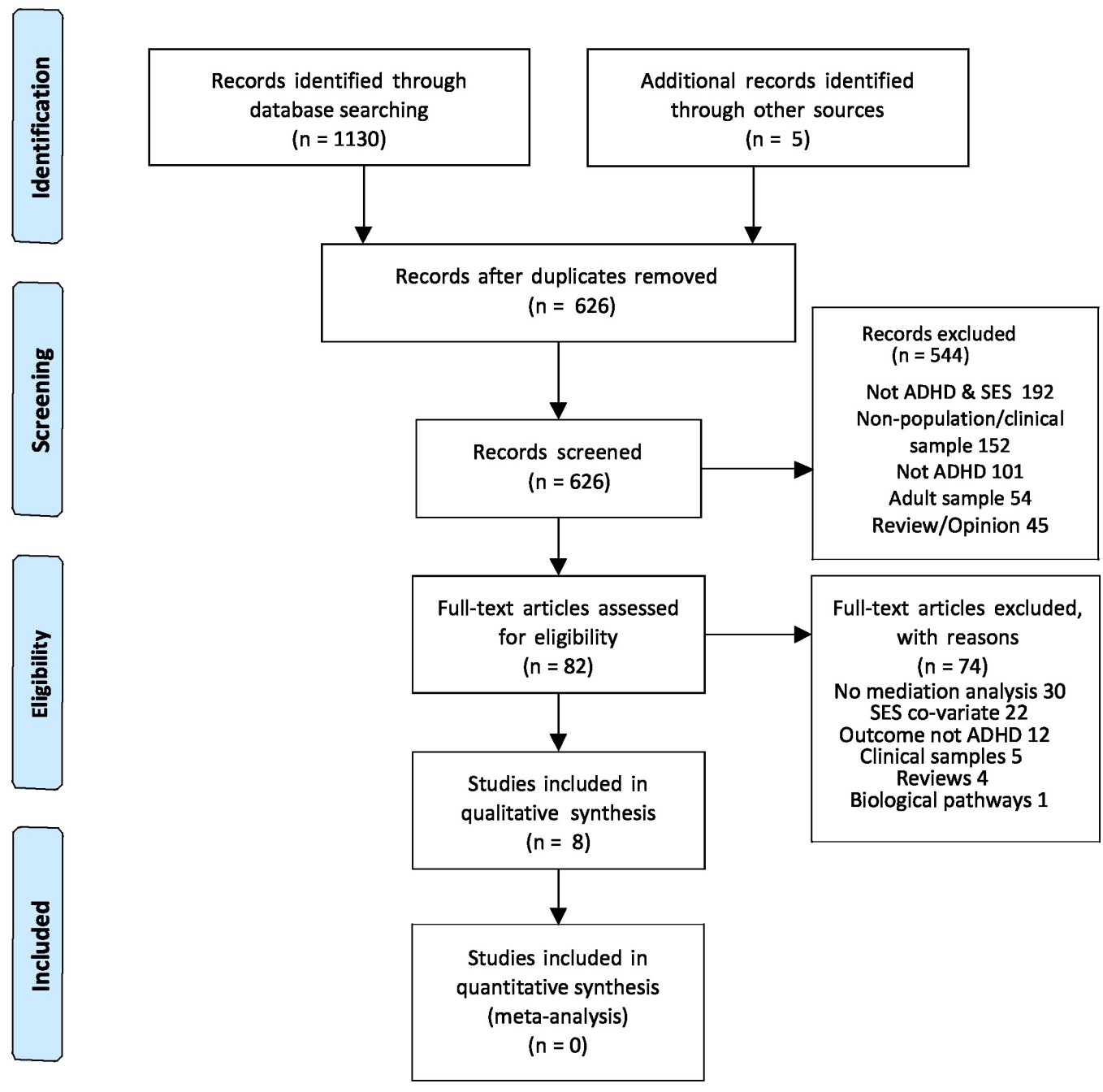

**Fig 2. Prisma flow chart [19].**

Reasons for excluding articles at the titles and abstract stage included: not ADHD & SES; non-general population/clinical sample; not ADHD; adult sample; review/opinion. We then assessed n = 82 full text articles and excluded n = 74 articles ([Fig 2]). Reasons for excluding full text articles included: no mediation analysis; SES was a co-variate not an exposure; the outcome was not ADHD or hyperactivity/inattention using standard psychometric tests; outcome was externalising behaviour including conduct problems or conduct disorder alone; the investigation was based upon a non-general population based sample; review article; the

investigated mediator was biochemical or physiological. Eight papers were identified that examined potential psychological, social or behavioural mediators of the relationship between household SES (exposure) and ADHD or hyperactivity/inattention in childhood/adolescence (outcome).

## Characteristics of included studies

The characteristics of included studies are summarized in Table 1. All included studies were conducted in North America (Canada and USA (2)) or Europe (France, Germany, Norway and UK (2)). five studies were cohort studies, two were cross-sectional studies and one was a case-control study. The populations studied were children aged 3 [23,24], aged 7 [12,25], aged 7–8 [26], aged 7–14 [27], aged 6–17 [28], and adolescents aged 17/18 [29].

Household SES was based upon maternal education when youngest child was 2 months old [24]; financial hardship when child was 0–2 (parent reported difficulty in affording heating, clothing, rent/mortgage, food and/or things for the study child) [12]; family household income at 17/18 [29]; and parental occupation when child was aged 4/5 [26]. Combined SES measures were used in four studies: parental education and household income pre-pregnancy [23]; parental education and household income when child was 7–14 years [27]; parental education, household income and child's health care insurance status when child was 6–17 years [28]; paternal education, maternal education, fathers' social class, mothers' social class at 9 months [25].

The measures of ADHD or hyperactivity/inattention using reliable and valid psychometric tests varied. Miller et al. [27] relied upon teacher-reported ADHD Rating Scale, Version VI (ADHD-RS-IV) [30], the Conners ADHD Rating Scale [31], and the Strengths and Difficulties Questionnaire (SDQ) [32]. Foulon et al. [23] used mother-reported SDQ questionnaires [33,34] and Schmiedeler et al. [26] used teacher-reported SDQ [33]. Meunier et al. [24] utilised the revised Ontario Child Health Study questionnaires [35] and recorded both parents' assessments. Russell AE et al. [12] used parent and teacher assessments based upon the Development and Well-Being Assessment (DAWBA) questionnaire [36]. Boe et al. [29] used adolescent-reported assessments based upon the WHO adult ADHD self-report ASRS scale [37]. Russell G et al. [25] and Nguyen et al. [28] relied upon parent-reporting of a health care provider diagnosis of ADHD.

The methods for investigating meditation varied between studies. Boe et al. [29], Miller et al. [27] and Nguyen et al. [28] conducted Structural Equation Modelling using Mplus version 7. Schmiedeler et al. [26] used Structural Equation Modelling in AMOS software along with Full Information Maximum Likelihood estimation for latent variable interactions. Foulon et al. [23] employed the MacArthur moderator-mediator path analysis approach [38]. Meunier et al. [24] used Baron and Kenny's traditional sequential framework for testing direct and indirect effects along with the multilevel modelling plus framework proposed by Edwards and Lambert [39] and Preacher et al. [40]. Two studies [12, 25] drew upon a mediation analysis method that adopts a products of coefficients approach [41].

Two of the included studies [25,29] did not adjust for covariates/confounders. All the other studies adjusted for the covariate, child's gender. Four studies [24,26–28] also adjusted for the covariate, child's age. In addition, Meunier et al. [24] adjusted for covariate, sibling gender composition. Schmiedeler et al. [26] adjusted for teacher reported ADHD at age 4 treating it as a covariate. Nguyen et al. [28] adjusted for the covariate, child's ethnicity, and a potential confounder, diagnosis of conduct problems.

## Risk of bias (RoB) assessment

Assessments of the quality of each included study's methodology are shown in S1 Table. None of the included studies were judged to have a low RoB. Four of the five cohort studies [12,23–

**Table 1. Characteristics of included studies.**

| Author Year Country | Study type; population; sample size; Attrition(%) | SES measure/s Age at SES measurement | ADHD measure Age at ADHD measurement Number (%) | Risk of bias | Mediators studied & method | Adjusted for covariates/ confounders | Direct & Indirect effects (95% CI) | Non-significant mediators |
|---|---|---|---|---|---|---|---|---|
| Boe et al. 2018 [29] Norway | Cross-sectional population-based study Working sample n = 9151 Attrition = 52.9% Mean age 17.47 years Standard deviation 0.84 years | Income to needs ratio = (Family household income adjusted for size)/ (60% median threshold for family household income adjusted for size) | Study participant-reported hyperactivity-Inattention continuous measure based on adult ADHD self-report ASRS scale Proportion with hyperactivity/ inattention not reported | High | Structural equation modelling Mplus Version 7.4 Mediator tested: Adolescents perceived economic status (poorer than others, equal to others, better than others) | No | Unstandardised β coefficients Direct Effect: $-0.208 - (-0.200) = -0.008$ (calculated from data) Indirect Effect: Via Perceived economic status $-0.200$ ($-0.253$, $-0.150$) Proportion of total effect $-0.208$ (95%CI $-0.315$, $-0.095$) explained by the indirect pathway ($0.200/0.200+0.008$) $= 9\%$ | |
| Foulon et al. 2015 France [23] | Cohort: EDEN project–women recruited in pregnancy from hospital maternity units in 2 large cities 1311 Attrition 31% | Pre-pregnancy monthly household income, paternal education, maternal education combined into a single measure | Mother-reported SDQ as continuous variable collected when child aged 3 yrs.–mean score = 3.5 Proportion with hyperactivity/ inattention not reported | Moderate | MacArthur moderator-mediator path analysis approach proposed by Kraemer et al. (2001) Mediators tested at 4 periods: Before pregnancy; prenatal/birth; infancy; toddlerhood. Foetal exposures Child's temperament Child's neurodevelopmental status Psychosocial environment | Child gender | Standardised β coefficients Direct Effect: $-0.18$ $p<0.05$ Indirect Effects: Via Breastfeeding duration ($0.25$ $p<0.5$, $-0.06$ $p<0.5$ Total indirect effect via this pathway $\beta = (0.25 \times -0.06) = -0.015$ Via Breastfeeding duration and Child neuro-developmental status (Combined score of fine motor score and language score at 2 years) ($0.25$ $p<0.5$, $0.08$ $p<0.5$, $-0.15$ $p<0.5$) Total indirect effect via this pathway $\beta = (0.25 \times 0.08 \times -0.15) = -0.003$ Via Maternal depression & anxiety combined assessed at 6 month of pregnancy and mother and Infant distress and dysregulation measured at 4-8-12 months ($-0.15$ $p<0.5$, $0.32$ $p<0.5$, $0.14$ $p<0.5$) Total indirect effect via this pathway was $\beta = (-0.15 \times 0.32 \times 0.14) = -0.007$ Proportion of total effects explained by the indirect pathways ($0.015+0.003+0.007$)/($0.015+0.003 +0.007$) + 0.18) $= 12\%$ | |

*(Continued)*

Table 1. (Continued)

| Author Year Country | Study type; population; sample size; Attrition(%) | SES measure/s Age at SES measurement | ADHD measure Age at ADHD measurement Number (%) | Risk of bias | Mediators studied & method | Adjusted for covariates/ confounders | Direct & Indirect effects (95% CI) | Non-significant mediators |
|---|---|---|---|---|---|---|---|---|
| Meunier et al. 2013 [24] Canada | Cohort: 920 children from 397 families with 2 or > children < 4 years Attrition = 20.8% | SES measure reported when youngest child was 2 months old: Number of years maternal education completed | Mean ADHD score measured by scale with well-established reliability and validity completed by both parents at 3 yrs. | Moderate | Baron and Kenny sequential framework and multilevel modelling plus framework proposed by Edwards and Lambert (2008). Mediators tested: Mother reported differential negativity Mother reported differential positivity Observed differential negativity in home Observed differential positivity measured in home | Age Child gender Sibling gender composition | Standardised β coefficients in separate single mediating risk factor models Positive and negative differential parenting included in sperate models Via Observed differential negativity Direct Effect: -0.17 (p<0.001) Indirect Effect: -0.016 (p < .05). Via Mother reported Differential positivity Direct effect: -0.16 (p<0.001) Indirect Effect: -0.015 (p < .05) Proportion of total effects explained by the indirect pathways Via Observed differential negativity (-0.016/(-0.17) + (-0.016)) = 9% Via mother-reported differential positivity (-0.015/(-0.015) + (0.16)) = 9% | Mother reported Differential negativity Direct Effect: -0.18 (p<0.001) Indirect Effect: 0.002 |
| Miller et al. 2016 [27] USA | Case-control study: n = 931 children 7–14 years Attrition N/A | A latent SES construct created from: Highest parental education, highest parental occupation, Family household income Mean age of chid = 9.3 years | Teacher ratings of hyperactivity/ inattention ADHD 521 Controls 335 Sub-threshold 75 | High | Structural Equation Modelling using Mplus 7.4 Mediators tested: Self-rated or spouse rated Paternal ADHD symptoms Self-rated or spouse rated Maternal ADHD symptoms | Age Child gender | Unstandardised β coefficients Direct Effect: -0.09 (p<0.05) Indirect Effects: Paternal ADHD symptoms -0.06 (p<0.001) Maternal ADHD symptoms -0.05 (p<0.01) Proportion of total effects explained by the indirect pathways (0.05 +0.06)/(0.05 +0.06) +0.11) = 55% | |

(Continued)

**Table 1.** (Continued)

| Author Year Country | Study type; population; sample size; Attrition(%) | SES measure/s Age at SES measurement | ADHD measure Age at ADHD measurement Number (%) | Risk of bias | Mediators studied & method | Adjusted for covariates/ confounders | Direct & Indirect effects (95% CI) | Non-significant mediators |
|---|---|---|---|---|---|---|---|---|
| Nguyen MN et al. 2019 [28] USA | Cross-sectional study: n = 65680 children 6–17 yrs. Mean age and standard deviation not reported Attrition N/A | SES was a latent variable made up of: Household income; Parent education; Parent employment; Child's health care insurance status | Parent report of a health care provider diagnosis of ADHD based upon 6560 (10% weighted) | High | Structural Equation Modelling using Mplus Mediators tested: Adverse Childhood Experiences (ACE) School engagement Neighbourhood safety Neighbourhood amenities | Age Child gender Child ethnicity Diagnosis of conduct problems. | Standardised β coefficients Direct Effect: No direct effect of SES on ADHD Indirect Effects: Total indirect effect (as reported in article) (β = − 0.03; p = 0.002) mostly via ACE and school engagement but also included neighbourhood safety in model Specific significant indirect effects SES to safety to school engaged to ACE to ADHD (β = − 0.08; p = < 0.001) SES to safety to ACE to school engaged to ADHD (β = − 0.01; p = <0.001) Proportion of total effects explained by the indirect pathways = 100% | Neighbourhood amenities |
| Russell AE et al. 2015 [12] UK | ALSPAC Birth cohort n = 8132 Attrition 45% | Main SES measure in analysis: Financial Hardship when child was 0–2. | Parent/carer and teachers reported ADHD based on DAWBA 7 years 172 (2.1%) | Moderate | Multiple mediation analysis method that adopts a products of coefficients approach using the products of coefficients approach (Preacher and Hayes, 2008) Mediators tested: Maternal involvement Paternal involvement Parental psychopathology (maternal depression) Child fizzy drinks consumption at 3 years Family adversity (Rutter score) | Child gender | Unstandardised β coefficients Direct Effect: 0.113(0.03,0.19) Indirect Effects: Via Mother involved 0.003(0.000–0.009) Via Partner involved 0.008 (0.001,0.015) Via Family adversity 0.028(0.012,0.050) Proportion of total effects explained by the indirect pathways (0.003+0.008+0.028)/(0.003+0.008 +0.028) + 0.113) = 26% | Maternal depression |

*(Continued)*

Table 1. (Continued)

| Author Year Country | Study type; population; sample size; Attrition(%) | SES measure/s Age at SES measurement | ADHD measure Age at ADHD measurement Number (%) | Risk of bias | Mediators studied & method | Adjusted for covariates/ confounders | Direct & Indirect effects (95% CI) | Non-significant mediators |
|---|---|---|---|---|---|---|---|---|
| Russell G et al. 2014 [25] UK | Cohort: UK-wide Millennium Cohort Study at 9 months 13305 Attrition 31.8% | SES Index based upon Fathers' social class, mothers' social class, paternal education, maternal education all measured at 9 months | Parent report of ADHD diagnosis by health professional at any time up to 7 years of age 187 (weighted % 1.5) | Moderate | Multiple mediation analysis method that adopts a products of coefficients approach using the products of coefficients approach (Preacher and Hayes, 2008) Mediators tested: Smoking in pregnancy Family conflict/distant parenting | No | Unstandardised β coefficients Direct Effect: 0.108 (0.003,0.205) p<0.05 Indirect Effect: Via Family conflict/distant parenting 0.045 (0.032,0.056) p<0.05 Proportion of total effects explained by the indirect pathway (0.45/(0.45+1.08) = 29% | Smoking in pregnancy 0.029 (-0.009, 0.069) |
| Schmiedeler et al. 2014 [26] Germany | Cohort: children attending mean age 4 yrs. n = 468 Attrition 49.4% Imputation used & final sample 924 | Wegener prestige scale for parental occupation assessed 4–5.25 yrs. States SES was assessed during kindergarten period (T1-T3), T1 mean age was 4 and T3 which was approximately 14 months later | Teacher report of SDQ at ages 7&8 | High | Structural Equation Modelling in AMOS software along with Full Information Maximum Likelihood estimation for latent variable interactions Mediators tested: Home learning environment TV exposure | Age Child gender Teacher-reported ADHD at mean age 4. Proportion with ADHD 5.6% (teachers' reports) | Unstandardised β coefficients Direct Effect: No direct effect of SES on hyperactivity/inattention Indirect effect: Via Home Learning Environment 0.08 p<0.05 Proportion of total effects explained by the indirect pathway = 100% | TV exposure |

25] had a moderate RoB. The single cohort study to be rated as high RoB [26] overcontrolled for SES. The other studies had a high RoB primarily because their design prevented assessments of the direction of causality as they were either cross-sectional studies [28,29] or a case control study [27].

Additional factors contributing to the RoB assessments included: not adjusting for covariates/confounders [25,29]; not clearly defining exposure [12]; non-representative sample [23,24,27]; high attrition rate (>20%) in all the cohort studies except for Schmiedeler et al. [26]; conflation between the exposure and the mediator [29] and overcontrolling for SES [26,28].

## Potential mediators of the SES-ADHD pathway in childhood and adolescence

Table 1 shows the significant and non-significant indirect effects that were investigated and Table 2 highlights the potential mediators that were investigated but did not meet the criteria for potential mediation as they were not associated with the exposure and/or the outcome.

**Table 2. Factors identified as not meeting criteria for potential mediation.**

| | Factors Identified as not meeting criteria for potential mediation |
|---|---|
| **Foulon et al. 2015 [23]** | **Mother's health-related behaviour** <br> Mean number of alcohol glasses/week (Measured at first trimester and third trimester during pregnancy) <br> Cannabis consumption (During pregnancy) <br> Maternal psychoactive drugs intake (When baby 4–12 months old and when baby 24 months old) <br> **Mother's psychological well being** <br> Maternal history of hospitalisation in psychiatry (pre-pregnancy) <br> Psychiatrist or psychologist consultation in the year before pregnancy(pre-pregnancy) <br> Number of psychiatrist or psychological consultations (When baby 4–12 months old and when baby 24 months old) <br> **Mother's characteristics** <br> Maternal age at first child <br> **Baby's characteristics at birth** <br> Birth weight, <br> Gestational age at delivery, <br> Apgar score at 5 minutes <br> Child required resuscitation at birth <br> **Baby/Infant/characteristics** <br> Baby unpredictable (When baby 4–12 months old) <br> Baby Inadaptable (When baby 4–12 months old) <br> Baby Dull (When baby 4–12 months old) <br> Child gross motor (When baby 24 months old) <br> **Family life** <br> Number of children with whom the child is cared (When baby 4–12 months old) <br> Number of stressful life events (When baby 4–12 months old) <br> Number of siblings (When baby 4–12 months old) <br> Parents living together (When baby 4–12 months old and when baby 24 months old) <br> Paternal involvement (When baby 4–12 months old) <br> Maternal child care (When baby 24 months old) |
| **Meunier et al. 2013 [24]** | **Family life** <br> Observed maternal differential positivity |
| **Russell AE et al. 2015 [12]** | **Mother's health-related behaviour** <br> Substance use (use of hard drugs or alcohol consumption of more than 3 glasses a day for more than 10 days) (At age 2–4) <br> **Infant/Toddler health-related behaviour** <br> Fizzy drinks/caffeine consumption at age 3 years <br> **Family life** <br> Partner cruelty (physical or emotional) (At age 2–4) |

## Parental psychological factors

**Maternal depression.** Foulon et al. [23] showed that one of the significant mediating pathways between pre-pregnancy household SES and inattention-hyperactivity at age 3 had two steps. The first step was via a combined perinatal maternal depression and anxiety factor that was based on scores from the Center for Epidemiologic Studies Depression Scale (CES-D) and the State Trait Inventory Anxiety (STAI) [42,43] (standardised β coefficient for indirect effect = -0.15). The second step was via impaired mother-child relationships based on postpartum depression symptoms [44] and infant's difficult temperament (standardised β coefficients for indirect effects = 0.32, 0.14). Thus, the total indirect effect via this pathway was β = (-0.15 X 0.32 X 0.14 = -0.007). However, other factors related to maternal psychological wellbeing were found not to meet the criteria for potential mediation. These factors included prepregnancy maternal history of psychiatric hospitalisation, psychiatrist or psychologist consultation in the year before pregnancy and number of psychiatrist or psychological consultations when the baby was 4–12 months old and 24 months old.

Russell AE et al. [12] used a different method for assessing mediation than Foulon et al. [23] and reported in their cohort study that maternal depression was not a significant mediating factor in the relationship between household SES and ADHD at age 7. In contrast to Foulon et al.'s 2-step pathway, Russell AE et al. [12] assessed maternal depression based upon a score of 13 or more on the Edinburgh Postnatal Depression Scale [45] measured when the child was 2 years and 9 months of age.

## Parental ADHD symptoms

Miller et al. [27] concluded that both paternal and maternal ADHD symptoms were significant mediating factors between household SES and ADHD among 7–14 year olds (β coefficient for indirect effect: Paternal ADHD symptoms −0.06 (p<0.001); Maternal ADHD symptoms −0.05 (p<0.01)). Their measure of parental ADHD was based upon self-reported/spouse-reported current and recalled ADHD symptoms using the Conners Adult ADHD rating scale (CAARS) ADHD index [46] and the Barkley Adult ADHD rating scale (BAARS) [47]. Parental ADHD is likely to precede both household SES and the child's ADHD. The temporal relationship of parental ADHD and its relationship with the child's ADHD suggests it is a confounder although it is theoretically possible for parental ADHD to be both a confounder and a mediator [48]. However, Miller et al. [27] acknowledge that parental ADHD "could statistically or mechanistically explain both social disadvantage (due to downward drift) and the child's ADHD" (p.2) which suggests treating parental ADHD as a mediator rather than a confounder is problematic.

## Parenting

Issues related to adverse parenting were investigated as potential mediating factors between household SES and childhood/adolescent ADHD or high scores for hyperactivity/inattention in four studies.

Russell G et al. [25] observed that family conflict/attachment, based upon the Child-Parent Relationship Scale [49] and measured when the child was 3 years old, was a significant mediating factor between household SES and ADHD when the child was 7 years old (β coefficient for indirect effect: 0.045 (95% CI (0.032,0.056)).

Russell AE et al. [12] identified that both maternal and paternal involvement/engagement in activities with their child at 6 years of age were significant mediators of the relationship between household SES and ADHD when the child was aged 7 (β coefficient for indirect effect: Mother involved 0.003 (95% CI 0.000–0.009); Partner involved 0.008 (95% CI 0.001,0.015)).

However, these findings did not echo the findings of Foulon et al. (2014) [23] who reported that both paternal involvement when the baby was 4–12 months old and maternal child care when the child was 24 months old did not meet the criteria for mediation between household SES and childhood inattention-hyperactivity when the child was aged 3.

**Differential parenting.** Differential parenting (favourable/unfavourable parental treatment of one sibling compared to another) was investigated as a potential mediating factor between household SES and ADHD when the child was aged 3 by Meunier et al. [24]. Observed differential negativity [50] and mother-reported differential positivity [51,52] were significant mediators of this relationship (β coefficients for indirect effect respectively: -0.016 (p < .05) and -0.015 (p < .05)) in separate single mediating risk factor models. Their study did however, report contradictory findings as mother-reported differential negativity [51,52] was not significant and observed maternal differential positivity [53] did not meet the criteria for potential mediation. These contradictory findings make the findings regarding parental differential negativity and positivity, in families with more than one child, [24] difficult to interpret.

## Family life

**Home Learning Environment.** Schmiedeler et al. [26] found that the relationship between household SES and hyperactivity-inattention when children were aged 7 and 8 was fully mediated by Home Learning Environment (β coefficient for indirect effect: -0.08 p<0.05) as the direct effect of household SES was non-significant. Home Learning Environment was measured by 11 questions including parents reading to their children, possessing books and daily newspapers, playing dice games with their children and owning a library card and visiting the library. High level of television viewing did not mediate the SES ADHD relationship. Television viewing (TV) focused on how many hours the child watched TV per day and how many hours the parent watched TV per day. However, Schmiedeler et al. [26] overcontrolled for SES by including as a covariate ADHD at a younger age.

**General family life characteristics.** Foulon et al. [23] reported that a number of other factors related to family life were not potential mediators between household SES and hyperactivity-inattention of 3 year old children as they did not meet the criteria for potential mediation. These factors included the number of children cared for when the child was 4–12 months old, number of siblings when the child was 4–12 months old and parents living together when the child was 4–12 months old and 24 months old.

## Family adversity including financial stress

Russell AE et al. [12] reported that family adversity when the child was 2–4 years was a significant mediating factor between household SES and ADHD at age 7 (β coefficient for indirect effect: 0.028 (95% CI (0.012,0.050)). Their family adversity index [54] was based on Rutter's original indicators of adversity [55] and included exposure to the following factors, lack of partner affection, partner cruelty (physical or emotional), family major problems, psychopathology of mother, substance use and trouble with the police. The authors also investigated partner cruelty and substance use on their own as potential mediators but neither of these factors on their own met the criteria for potential mediation. Stressful life events are more commonly experienced by low SES households [56]. However, Foulon et al. [23] reported that the number of stressful life events when the baby was 4–12 months old did not meet the criteria for potential mediation.

Nguyen et al. [28] reported that household SES had no direct effect on ADHD when the child was aged between 6 and 17 years after accounting for indirect effects of multiple mediators (Adverse Childhood Experiences (ACEs), school engagement, neighbourhood safety and

neighbourhood amenities). Household SES was a latent variable in their Structural Equation Model made up of household income, parent education, parent employment and child's health care insurance status. Nonetheless, household SES did have an indirect effect on ADHD when the child was aged between 6 and 17 years mostly via ACEs and school engagement (Total indirect effect standardised β coefficient = − 0.03; p = 0.002). ACE was represented by nine hardships including financial stress, having lived with divorced/separated parent, lived with a parent who died or served time in jail, having witnessed domestic violence, having been a victim or witnessed violence in the neighbourhood, having lived with someone with mental health problems or a substance use problems and having experienced racial discrimination. School engagement focused on caring about doing well at school and doing all the required homework. Neighbourhood safety focussed on feeling safe in their neighbourhood and feeling safe at school. Neighbourhood amenities focussed on having pavements, a park, a recreational centre or a library. However, these authors over controlled for SES in the mediation analysis because ACEs included financial stress which is a recognised measure of SES. [12] This is likely to diminish the direct effect of SES.

**Economic factors.** Boe et al. [29] concluded that adolescents' perceived economic status at mean age 17.5 years was a significant mediating factor between household SES based upon adjusted family household income and adolescent hyperactivity/inattention (β coefficient for indirect effect: -0.200 (95% CI (-0.253, -0.150)). However, adolescents' perceived economic status and household parental SES are likely to be conflated. Moreover, the temporal relationship between adolescent perceived economic status and ADHD in the cross-sectional study by Boe et al. [29] is unclear. The assumption in this study is that adolescents' perceived economic status precedes hyperactivity/inattention measured once at 17.5 years; however, the natural history of hyperactivity/inattention which commonly starts in early childhood suggests the reverse i.e. the condition in these adolescents would precede their perception of economic status.

## Behavioural factors that mediate the relationship between household SES and childhood/adolescent ADHD or hyperactivity/inattention

Breastfeeding duration during the period (4-8-12 months) was reported by Foulon et al. [23] to mediate the relationship between pre-pregnancy household SES and childhood inattention/hyperactivity at age three via two pathways. The first one-step pathway was via breastfeeding duration only (standardised β coefficients for indirect effect = 0.25 and -0.06; Total indirect effect via this pathway β = -0.015). The second two-step pathway was via breastfeeding duration during the period (4-8-12 months) (Step 1) (standardised β coefficients for indirect effect = 0.25,) followed by child neuro-developmental status (combined score of fine motor and language score at 2 years) (Step 2) (standardised β coefficient for indirect effects = 0.08, -0.15). Thus, the total indirect effect via this pathway β = (0.25 X 0.08 X -0.15) = -0.003.

Two studies [23,25] reported that smoking during pregnancy was not a significant mediator of the SES-ADHD relationship. Foulon et al. [23] also reported that other health-related behaviours did not meet the criteria for potential mediation such as mean number of alcohol glasses/week (measured at first and third trimester during pregnancy), cannabis consumption during pregnancy, maternal psychoactive drugs intake when the baby was 4–12 months old and 24 months old. Moreover, Russell AE et al. (2015) [12] found that substance use (use of hard drugs or alcohol consumption of more than 3 glasses a day for more than 10 days) during the period that the baby was aged 2–4 did not meet the criteria for potential mediation. This study also found that the child's consumption of fizzy drinks/caffeine when it was 3 years old did not meet the criteria for potential mediation.

## Socio-biological characteristics of mother and baby that mediate the relationship between household SES and childhood/adolescent ADHD or hyperactivity/inattention

As highlighted above, Foulon et al [23] observed a two-step pathway between household SES and childhood inattention-hyperactivity that focussed on breastfeeding duration (Step 1) and child neuro-developmental status (combined score of fine motor and language score at 2 years). However, other developmental characteristics investigated by the authors did not meet the criteria for potential mediation including the baby being unpredictable when it was 4–12 months old, the baby being unadaptable when it was 4–12 months old, the baby being dull when it was 4–12 months old and gross motor score when the child was 24 months old [57].

These authors also reported that mother's age at birth was not a significant mediator of the relationship between household SES and childhood inattention-hyperactivity. Additionally, mother's age when she had her first child and baby's birth weight, gestational age, Apgar Score at 5 minutes and requiring resuscitation at birth did not meet the criteria for potential mediation.

## Direct effects of household SES on child/adolescent ADHD or hyperactivity/inattention

The proportion of the total effects of household SES on child/adolescent ADHD or hyperactivity/inattention that was mediated varied between studies. The proportions mediated were smaller in the four cohort studies we assessed as having a moderate RoB than in the studies we assessed as having a high ROB. In the moderate RoB cohort studies, mediation accounted for 9% of the total SES effect in both of the separate models of Meunier et al. [24], 12% in Foulon et al. [23], 26% in Russell AE et al. [12] and 29% in Russell G et al. [25]. In the high RoB studies, mediation accounted for 100% of the total SES effect in two studies [26,28], 96% in one [29] and 55% in another [27] (Table 1).

Meunier et. [24] reported direct effects of household SES on hyperactivity/attention among 3 year old children of -0.17 (p<0.001) and -0.16 (p<0.001).

Foulon et al. [23] reported a direct effect of household SES of -0.18 (standardised β coefficient) (p<0.05) on ADHD among 3 year olds.

Russell AE et al. [12] found a direct effect of household SES on ADHD among 7 year olds of 0.113 (95% CI (0.03,0.19)) after accounting for the effects of maternal depression, family adversity, mother involvement and father involvement as potential mediators.

Russell G et al. [25] reported a direct effect of household SES on ADHD among 7 year olds of 0.108 (95% CI (0.003,0.205).

Regarding the four studies we assessed as having a high RoB, Boe et al. [29] do not report the direct effect of objective household SES. However, calculation from study data identified a small direct effect of household SES (0.008) on self-reported hyperactivity/inattention among adolescents aged 17/18 years. The only case-control study we included in this review [27] reported a relatively small direct effect of household SES on teacher-rated hyperactivity/inattention among 7–14 year olds of -0.09 (p<0.05).

Two studies we assessed as having a high RoB [26,28] reported no direct effect of household SES on ADHD or hyperactivity/inattention but both of these studies overcontrolled for household SES. Nguyen et al. [28] focussed on parent-reported health care provider diagnosis of ADHD when the child was aged between 6 and 17 years. Schmiedeler et al. [26] focussed on teacher-reported hyperactivity/inattention among 7–8 year olds.

## Discussion

Our review shows that evidence for mediation of the pathway between household SES and ADHD in childhood/adolescence is sparse and under-researched. We only identified eight studies that met the inclusion criteria which examined a range of psychological, social and behavioural risk factors as potential mediators. There were no studies of child populations outside North America and northern Europe. We synthesised the quantitative data using the SWiM guidelines for narrative synthesis without meta-analysis as diversity of mediators studied, study designs, population samples, exposure and outcome measures, and study methods precluded meta-analysis. For example, maternal depression was measured differently at different times in the foetal/infant life course and potential mediation was tested by different methods.

### Main findings

When indirect effects of mediators were accounted for in the four cohort studies we assessed as having a moderate RoB, the direct effects of household SES on ADHD were robust with proportions mediated less than 30%. These results are likely to have greater validity than those from the high RoB studies which reported mediation of all or a high proportion of the household SES effect.

The review found supporting evidence for mediation of the SES-ADHD pathway by parenting behaviours, including parental conflict/attachment [25], parental engagement [12], parental differential negativity and positivity [24], and Home Learning Environment [26]. Breast feeding was the only health behaviour shown to mediate the SES-ADHD pathway. [23] One study [12] reported mediation by maternal anxiety and depression, known risk factors for ADHD in children [58]; however, a study using a different methodology did not support mediation. [23] Mediation by Adverse Childhood Experiences (ACEs), which are strongly correlated with SES [59], was reported by two studies [12,28] but a further study did not support mediation [23].

The review found no evidence of a mediating role for smoking in pregnancy, alcohol and/or cannabis consumption during pregnancy, maternal substance abuse of hard or psychoactive drugs during the child's early years and child's consumption of fizzy drinks. Known socio-biological risk factors for ADHD in children [11], including maternal age at the child's birth and the birth of her first child, the baby's birth weight, gestational age at delivery, and Apgar score at 5 minutes, all failed to meet the criteria for potential mediation [23].

**Methodological limitations of the studies included in this review.** The included papers had substantial methodological limitations. An essential prerequisite of mediation analysis is that the exposure precedes the outcome and potential mediators temporally lie between the exposure and the outcome. These temporal relationships are verifiable in longitudinal studies but are more difficult to verify in cross-sectional and case-control studies. Nguyen et al. [28] acknowledge this limitation but suggest, without supporting evidence, that using SEM in their study allows directionality of the variables to be examined in cross-sectional data. The temporal relationship between ADHD and both adolescent perceived economic status and parental ADHD makes interpretation of the findings of respectively Boe et al. [29] Miller et al. [27] problematic.

All the included studies had limitations related to study samples. Foulon et al. [23] recruited pregnant women from hospital maternity units in two large French cities and it is not clear if this hospital-based sample is representative of the target population. The cohort recruited by Schmiedeler et al. [26] was embedded in a national longitudinal study but the representativeness of the sub-sample is unclear. Meunier et al. [24] excluded families with a single child from

their cohort sample and Miller et al. [27] recruited both their cases and controls using community mailout lists and public advertisements which were unlikely to be representative of the target population.

Non-participation rates exceeded 50% in three studies [24,28,29] and were not reported in four studies [12,25–27] potentially introducing significant selection bias. Attrition is a universal problem in cohort studies. Sample weighting and imputation can be used to minimise the bias associated with differential loss to follow by socially related factors. [60] Attrition rates in all the included cohorts exceeded 20% and only one study [25] reported using weighting for analysis of the association of SES with ADHD but used unweighted data in the mediation analysis. The authors justify the use of unweighted data citing evidence that unweighted regression models are often robust in large datasets [61].

Two studies [25,29] did not adjust for covariates/confounders introducing a source of potential bias as confounders may falsely obscure or accentuate the exposure/outcome relationship. Two studies introduced socially related variables which are likely to have resulted in over-controlling for SES and reducing the total and direct effects of SES on ADHD [26,28].

**Validity.** Mediation by parenting behaviour reported by two moderate RoB studies [12,25] are more likely to be valid finding. The findings that maternal health-related behaviours in pregnancy or in the child's first three years of life and peri-natal sociobiological factors either did not mediate the SES- ADHD pathway or failed to meet the criteria for mediation are likely to be valid as they are reported by moderate RoB studies [12,23]. The finding that the Home Learning Environment mediates the SES-ADHD pathway [26] is open to question as the study carries a high RoB and it is not clear if the measure of Home Learning Environment has been properly validated. Failure to include evidence of the validity of the methods used to derive adolescents' perceived SES in Boe et al [29] and school engagement and neighborhood safety/amenities in Nguyen et al [28], also threatens the validity of these study results.

**Methods of mediation analysis.** All the included studies employed recognized, valid mediation analysis methods; however, these methods have limitations as they do not take account of models with interactions and non-linearities [62]. As a consequence, the methods used in the included papers may be subject to bias due to the mediator being affected by the exposure which, in turn, confounds the relationship between the mediator and the outcome—the exposure-induced mediator-outcome confounder effect. This effect becomes more likely the longer the period between the exposure and the outcome [62] suggesting that the cohort studies included in the review are susceptible to this limitation.

## Strengths and limitations of the systematic review

The review protocol was registered in the PROSPERO review registry following revisions. We

- employed a robust method of risk of bias assessment modified to include potential sources of bias specific to studies of mediation

- only included studies in which the outcome was ADHD or hyperactivity/inattention as externalizing behaviour in standard psychometric tests includes conduct disorder which has a different relationship with SES

- followed the methodologically robust SWiM protocol [20] for narrative synthesis of the included studies

Grey literature databases were not included in the search strategy and additional studies may have been missed. However, this is a narrow field of academic interest and studies meeting the inclusion criteria are very likely to have been published in international peer-reviewed

journals. Publication bias may have been increased by our decision not to approach study authors and limit the search to English language publication. Exclusion of studies examining potential biochemical and physiological mediators from the review may have limited assessment of the full range of mediators reported in the literature; however, we sought to identify modifiable psychological, social and behavioural mediators and factors such as brain morphology are unlikely to be modifiable.

### Implications for future research and policy development

Future research should be based on methodologically robust longitudinal studies with comparable measures of psychological, social and behavioural factors which meet the criteria for mediation. Maternal mental health, family adversity, parenting and health-related behaviours warrant further research. Studies will need to be sufficient in number and quality to enable meta-analysis to estimate robust pooled indirect effects of potential mediators on the SES-ADHD pathway. Studies should employ the most advanced mediation analysis methods which account for potential exposure-induced mediator outcome confounding [62]. More robust, methodologically sound research is important for future policy development aimed at minimising the link between low household SES in early childhood and later ADHD.

### Conclusions

A range of psychological, social and behavioural risk factors have been studied as potential mediators of the SES-ADHD pathway; however, reliable conclusions on their effects on the pathway are limited by the small number of studies combined with the moderate to high risk of bias in these studies and diversity of study design, mediators studied and measurement SES and ADHD. However, no evidence of effect is not the same as evidence of no effect. Hence, we propose maternal mental health, family adversity, parenting and health-related behaviours warrant further research.

### Supporting information

**S1 Table. Risk of bias (quality) assessment.**
(DOCX)

**S1 File. Prospero systematic review protocol (registration no. CRD42020182832).**
(DOCX)

**S2 File. PRISMA-2009 checklist.**
(DOC)

**S3 File. Search strategies for electronic databases.**
(DOCX)

### Acknowledgments

We are most grateful to Mrs Samantha Johnson, Senior Health Science Librarian, University of Warwick for her expert advice and assistance in preparing the search strategy and running the literature searches. We are also grateful to Claire New for administrative support.

### Author Contributions

**Conceptualization:** Wolfgang A. Markham, Nicholas Spencer.

**Data curation:** Nicholas Spencer.

**Formal analysis:** Wolfgang A. Markham, Nicholas Spencer.

**Investigation:** Wolfgang A. Markham, Nicholas Spencer.

**Writing – original draft:** Wolfgang A. Markham.

**Writing – review & editing:** Wolfgang A. Markham, Nicholas Spencer.

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
