## [Decision Letter · Decision Letter 0]

12 Aug 2021

PONE-D-21-03193

Factors that mediate the relationships between household socio-economic status and childhood Attention Deficit Hyperactivity Disorder (ADHD) in children and adolescents: A systematic review

PLOS ONE

Dear Dr. Markham,

Thank you for submitting your manuscript to PLOS ONE. After careful consideration, we feel that it has merit but does not fully meet PLOS ONE’s publication criteria as it currently stands. Therefore, we invite you to submit a revised version of the manuscript that addresses the points raised during the review process.

The reviewers identified some major issues those need to be fixed before taking final decision.

We look forward to receiving your revised manuscript.

Kind regards,

Enamul Kabir

Academic Editor

PLOS ONE

Journal Requirements:

4. Please upload a new copy of Figure 2 as the detail is not clear. Please follow the link for more information: " ext-link-type="uri" xlink:type="simple">https://blogs.plos.org/plos/2019/06/looking-good-tips-for-creating-your-plos-figures-graphics/"
" ext-link-type="uri" xlink:type="simple">https://blogs.plos.org/plos/2019/06/looking-good-tips-for-creating-your-plos-figures-graphics/".

Reviewers' comments:

Reviewer's Responses to Questions

**Comments to the Author**

1. Is the manuscript technically sound, and do the data support the conclusions?

Reviewer #1: Yes

Reviewer #2: Partly

2. Has the statistical analysis been performed appropriately and rigorously? 

Reviewer #1: N/A

Reviewer #2: I Don't Know

3. Have the authors made all data underlying the findings in their manuscript fully available?

Reviewer #1: Yes

Reviewer #2: No

4. Is the manuscript presented in an intelligible fashion and written in standard English?

Reviewer #1: Yes

Reviewer #2: Yes

5. Review Comments to the Author

Reviewer #1: Comments

Overall

1. Follow PLOS author’s guideline strictly

2. Content wise, authors studied novel title and did synthesis as per available literatures

Specific

1. Authors are advised to keep reference after full stop eg. Instead of “case ascertainment remains unclear [3] [4].” write like “case ascertainment remains unclear. [3][4]”. For authors ease, I suggest authors to use referencing with Mendeley plugin of PLOS One format, which cut down the referencing errors! Additionally, authors kept NLM abbreviations of some journals but not in all cases, so please check and keep them as per guideline or Plugin for uniformity!

2. Though common abbreviation are understandable to readers, but in writing it is advised to keep full form of any abbreviation in body of manuscript upon their first use. Eg. “ADHD”, “USA”, elaborate them upon first use in section Introduction!

3. In result and discussion, some write up came as redundant and repeated, so advised to proof read to avoid unnecessary repetition of same

Reviewer #2: Comments:

Minor issue is:

1. Introduction: starting from line 58, you have wrote about the prevalence of ADHD in USA since 2011, your study time is 2020, why not discus (look) after 2011.

2. Results: starting line 166, you have screened n = 626 titles and abstracts for eligibility and n = 82 are assessed and 72 are exclude, what about the rest or the remaining titles and abstracts.

3. Line 168, you have assessed n = 82 titles and abstracts, and line 180 you have categorized by study type: Five studies were cohort studies, two were cross-sectional studies and one was a case-control study. When we add the type it is 8, how you categorized?

4. The figures are not visible.

5. Your article is a very lengthy (51 pages). It should be revised and shortened substantially.

6. Check your manuscript carefully with PLOS One author guidelines.

6. PLOS authors have the option to publish the peer review history of their article (what does this mean?). If published, this will include your full peer review and any attached files.

Reviewer #1: No

Reviewer #2: No

---

## [Author Response · Author response to Decision Letter 0]

21 Nov 2021

Response to reviewers 

Comment by Reviewer 

Reviewer 1 

1. Follow PLOS author’s guideline strictly 

We have now checked the manuscript carefully with PLOS One author guidelines. Thank you for pointing this out.

2. Content wise, authors studied novel title and did synthesis as per available literatures 

Thank you for your positive comments.

1. Authors are advised to keep reference after full stop eg. Instead of “case ascertainment remains unclear [3] [4].” write like “case ascertainment remains unclear. [3][4]”. For authors ease, I suggest authors to use referencing with Mendeley plugin of PLOS One format, which cut down the referencing errors! 

We have changed this according to the PLOS One author guidelines. Thank you for pointing this out.

Additionally, authors kept NLM abbreviations of some journals but not in all cases, so please check and keep them as per guideline or Plugin for uniformity! 

We have amended this. Thank you for pointing this out. 

2. Though common abbreviation are understandable to readers, but in writing it is advised to keep full form of any abbreviation in body of manuscript upon their first use. Eg. “ADHD”, “USA”, elaborate them upon first use in section Introduction! 

We have made the changes and kept the full form of any abbreviation in the body of the manuscript upon its first use. Thank you for pointing this out.

3. In result and discussion, some write up came as redundant and repeated, so advised to proof read to avoid unnecessary repetition of same We have shortened the paper and reduced the repetition. 

The revised Results section covers the narrative synthesis according to the SWiM guidelines. We think following the SWiM guidelines is important but in order to do this requires additional text and is therefore relatively wordy. However, we have removed most of the reiteration of results that was in the Discussion section in our originally submitted paper. We have also shortened the section entitled “Direct effects of Household SES on child/adolescent ADHD or hyperactivity/inattention”. We have consequently shortened the paper and the resubmitted version has 769 fewer words. We hope that the shorter version is an improvement. Thank you for the suggestion.

Reviewer 2 

Minor issue is:

1. Introduction: starting from line 58, you have wrote about the prevalence of ADHD in USA since 2011, your study time is 2020, why not discus (look) after 2011.

We have included the date of the meta-regression which is 2017 and is therefore much more recent than 2011 . This meta-regression included studies from a wide variety of countries. 

A meta-regression, conducted in 2017, of over 100 studies spanning the globe identified that prevalence rates hover around 5%.

2. Results: starting line 166, you have screened n = 626 titles and abstracts for eligibility and n = 82 are assessed and 72 are exclude, what about the rest or the remaining titles and abstracts.

The reviewer is incorrect in that we stated that we excluded 74 articles rather than 72 as stated by the reviewer. We hope that this is clearer in the new Prisma flow chart that we have uploaded.

We have also amended this section and hopefully made it clearer 

After removing duplicates we screened n=626 titles and abstracts for eligibility. Reasons for excluding articles at the titles and abstract stage included: not ADHD SES; non-general population/clinical sample; not ADHD; adult sample; review/opinion. We then assessed n=82 full text articles and excluded n=74 articles (Fig. 2). Reasons for excluding full text articles included: no mediation analysis; SES was a co-variate not an exposure; the outcome was not ADHD or hyperactivity/inattention using standard psychometric tests; outcome was externalising behaviour including conduct problems or conduct disorder alone; the investigation was based upon a non-general population based sample; review article; the investigated mediator was biochemical or physiological. 

3. Line 168, you have assessed n = 82 titles and abstracts, and line 180 you have categorized by study type: Five studies were cohort studies, two were cross-sectional studies and one was a case-control study. When we add the type it is 8, how you categorized? 

We think that it is possible that as a consequence of reading that we excluded 72 full text articles rather than 74 full text articles the reviewer potentially makes the assumption that we included 10 articles rather than 8. 

We have made this sentence clearer by stating 

Of the eight articles we included, five studies were cohort studies, two were cross-sectional studies and one was a case-control study.

4. The figures are not visible. We are not sure why the Figures are not visible. However, as recommended we have uploaded a different Fig 2 which we hope is visible.

5. Your article is a very lengthy (51 pages). It should be revised and shortened substantially. The revised Results section covers the narrative synthesis according to the SWiM guidelines. We think following the SWiM guidelines is important but in order to do this requires additional text and is therefore relatively wordy. However, we have removed most of the reiteration of results that was in the Discussion section in our originally submitted paper. We have also shortened the section entitled “Direct effects of Household SES on child/adolescent ADHD or hyperactivity/inattention”. We have consequently shortened the paper and the resubmitted version has 769 fewer words. We hope that the shorter version is an improvement. Thank you for the suggestion. 

6. Check your manuscript carefully with PLOS One author guidelines. 

We have now checked the manuscript carefully with PLOS One author guidelines. Thank you for pointing this out.

---

## [Decision Letter · Decision Letter 1]

11 Jan 2022

Factors that mediate the relationships between household socio-economic status and childhood Attention Deficit Hyperactivity Disorder (ADHD) in children and adolescents: A systematic review

PONE-D-21-03193R1

Dear Dr. Markham,

We’re pleased to inform you that your manuscript has been judged scientifically suitable for publication and will be formally accepted for publication once it meets all outstanding technical requirements.

Kind regards,

Enamul Kabir

Academic Editor

PLOS ONE

Additional Editor Comments (optional):

Reviewers' comments:

Reviewer's Responses to Questions

**Comments to the Author**

1. If the authors have adequately addressed your comments raised in a previous round of review and you feel that this manuscript is now acceptable for publication, you may indicate that here to bypass the “Comments to the Author” section, enter your conflict of interest statement in the “Confidential to Editor” section, and submit your "Accept" recommendation.

Reviewer #1: All comments have been addressed

Reviewer #2: All comments have been addressed

2. Is the manuscript technically sound, and do the data support the conclusions?

Reviewer #1: Yes

Reviewer #2: Yes

3. Has the statistical analysis been performed appropriately and rigorously? 

Reviewer #1: (No Response)

Reviewer #2: I Don't Know

4. Have the authors made all data underlying the findings in their manuscript fully available?

Reviewer #1: Yes

Reviewer #2: No

5. Is the manuscript presented in an intelligible fashion and written in standard English?

Reviewer #1: Yes

Reviewer #2: Yes

6. Review Comments to the Author

Reviewer #1: Authors have responded on the comments in appropriate ways. Based on the methods provided, research is conducted appropriately.

Reviewer #2: I have checked your revised manuscript and your response to my comments. All of my comments are addressed by the author.

7. PLOS authors have the option to publish the peer review history of their article (what does this mean?). If published, this will include your full peer review and any attached files.

Reviewer #1: No

Reviewer #2: No

---

## [Editor Report · Acceptance letter]

21 Feb 2022

PONE-D-21-03193R1 

Factors that mediate the relationships between household socio-economic status and childhood Attention Deficit Hyperactivity Disorder (ADHD) in children and adolescents: A systematic review 

Dear Dr. Markham:

I'm pleased to inform you that your manuscript has been deemed suitable for publication in PLOS ONE. Congratulations! Your manuscript is now with our production department. 

Kind regards, 

on behalf of

Dr. Enamul Kabir 

Academic Editor

PLOS ONE